# Crosstalk of Inflammation and Coagulation in *Bothrops* Snakebite Envenoming: Endogenous Signaling Pathways and Pathophysiology

**DOI:** 10.3390/ijms241411508

**Published:** 2023-07-15

**Authors:** Joeliton S. Cavalcante, Denis Emanuel Garcia de Almeida, Norival A. Santos-Filho, Marco Aurélio Sartim, Amanda de Almeida Baldo, Lisele Brasileiro, Polianna L. Albuquerque, Sâmella S. Oliveira, Jacqueline Almeida Gonçalves Sachett, Wuelton Marcelo Monteiro, Rui Seabra Ferreira

**Affiliations:** 1Graduate Program in Tropical Diseases, Botucatu Medical School (FMB), São Paulo State University (UNESP—Univ Estadual Paulista), Botucatu 18618-687, São Paulo, Brazil; joeliton.cavalcante@unesp.br; 2Department of Bioprocess and Biotechnology, School of Agriculture, Agronomic Sciences School, São Paulo State University (UNESP—Univ Estadual Paulista), Botucatu 18618-687, São Paulo, Brazil; denis.garcia@unesp.br; 3Institute of Chemistry, São Paulo State University (UNESP—Univ Estadual Paulista), Araraquara 14800-900, São Paulo, Brazil; norival.santos-filho@unesp.br; 4Laboratory of Bioprospection, University Nilton Lins, Manaus 69058-030, Amazonas, Brazil; marcosartim@hotmail.com; 5Research & Development Department, Nilton Lins Foundation, Manaus 69058-030, Amazonas, Brazil; jac.sachett@gmail.com (J.A.G.S.); wueltonmm@gmail.com (W.M.M.); 6Graduate Program in Tropical Medicine, Department of Research at Fundação de Medicina Tropical Dr. Heitor Vieira Dourado, Amazonas State University, Manaus 69850-000, Amazonas, Brazil; liselebrasileiro@hotmail.com; 7Institute of Biosciences, São Paulo State University (UNESP—Univ Estadual Paulista), Botucatu 18618-687, São Paulo, Brazil; aa.baldo@unesp.br; 8Toxicological Information and Assistance Center, Instituto Doutor Jose Frota Hospital, Fortaleza 60025-061, Ceará, Brazil; pollylemos78@gmail.com; 9Faculty of Medicine, University of Fortaleza, Fortaleza 60430-140, Ceará, Brazil; 10Research Management, Hospital Foundation of Hematology and Hemotherapy of Amazonas, Manaus 69050-001, Amazonas, Brazil; oliveira.samella@gmail.com; 11Center for Translational Science and Development of Biopharmaceuticals FAPESP/CEVAP-UNESP, Botucatu 18610-307, São Paulo, Brazil; 12Center for the Study of Venoms and Venomous Animals (CEVAP), São Paulo State University (UNESP—Univ Estadual Paulista), Botucatu 18610-307, São Paulo, Brazil

**Keywords:** complement, hemostasis, inflammation, neutrophil, platelet, snake venom, thromboinflammation

## Abstract

Snakebite envenoming represents a major health problem in tropical and subtropical countries. Considering the elevated number of accidents and high morbidity and mortality rates, the World Health Organization reclassified this disease to category A of neglected diseases. In Latin America, *Bothrops* genus snakes are mainly responsible for snakebites in humans, whose pathophysiology is characterized by local and systemic inflammatory and degradative processes, triggering prothrombotic and hemorrhagic events, which lead to various complications, organ damage, tissue loss, amputations, and death. The activation of the multicellular blood system, hemostatic alterations, and activation of the inflammatory response are all well-documented in *Bothrops* envenomings. However, the interface between inflammation and coagulation is still a neglected issue in the toxinology field. Thromboinflammatory pathways can play a significant role in some of the major complications of snakebite envenoming, such as stroke, venous thromboembolism, and acute kidney injury. In addition to exacerbating inflammation and cell interactions that trigger vaso-occlusion, ischemia–reperfusion processes, and, eventually, organic damage and necrosis. In this review, we discuss the role of inflammatory pathways in modulating coagulation and inducing platelet and leukocyte activation, as well as the inflammatory production mediators and induction of innate immune responses, among other mechanisms that are altered by *Bothrops* venoms.

## 1. Introduction

Snakebite envenoming remains the main neglected disease in tropical and subtropical countries due to morbidity and mortality high rates worldwide [1,2]. In Latin America and the Caribbean, at least 137,000 snakebites and 3400 deaths occur annually [3], and *Bothrops* represents the major snake genus that causes snakebite envenoming, sometimes resulting in tissue loss/limb amputations and permanent disability in humans [4]. Currently, 36 species of *Bothrops* have been cataloged, and the species display great variability, mainly regarding their color and size patterns, as well as the compositions and biological activities of their venoms [5,6,7]. *Bothrops asper* is a major snake that is responsible for human envenoming in Central America, while in South America *B. atrox*, *B. erythromelas*, *B. jararaca*, *B. jararacussu*, and *B. moojeni* are mainly responsible for snakebite envenoming in the region [8]. 

*Bothrops* venoms are mostly composed of zinc-dependent metalloproteases (SVMPs), phospholipases A_2_ (PLA_2_s), and serine proteases (SVSPs), which in situations of human envenoming, trigger a wide spectrum of pathophysiological manifestations [7,9,10,11,12,13,14,15], such as pain, edema, blistering, myonecrosis, vascular injury, ischemia, necrosis, blood incoagulability, oxidative stress, bleeding, among others [16,17,18]. Although the wide variability in the venom proteome of *Bothrops* is recognized, this pattern of local and systemic clinical manifestations is replicated among envenomed patients by species of this genus, which suggests the presence of a shared mechanism of action, which can lead to complications such as intracranial hemorrhage, acute kidney injury, compartment syndromes, and amputations [8,19,20,21,22,23,24,25].

The coagulation disorders observed in *Bothrops* snakebites are induced by hemostatically active components, such as thrombin-like, anti-, and procoagulant toxins, which generate a consumption coagulopathy, and result in bleeding and thrombotic events [26]. In addition, thrombocytopenia is also observed among cases of *Bothrops* envenoming [27,28,29]. Although the toxins characterization from these venoms that act on platelets is wide, the involvement of platelets in the pathogenesis of *Bothrops* envenoming has received less attention, neglecting the role of these cells in other phenomena such as inflammation [30].

The in-depth knowledge of the differences between the coagulation cascade and multicellular systems in *Bothrops* envenoming makes it feasible to understand the isolated role of these systems in the pathology, as well as the complete pathophysiological scenario. While the inflammation caused by *Bothrops* venoms is characterized by pain, edema formation, erythema, and cellular infiltrate, the hemostasis, platelets, leukocytes, complement system, and inflammation are tightly interlinked processes, and platelets are often the cell type that binds these processes together [16,30,31]. In this context, the term “thromboinflammation” has been used to describe the activation of the cascading blood systems, as well as the activation of the multicellular blood system [32].

In the current study “thromboinflammation” in *Bothrops* envenoming was considered to be the activation of integrated processes, which includes classical coagulation, clot formation, and immune responses elicited during *Bothrops* envenoming [30,33,34,35]. Thus, the thromboinflammation process was recognized as a broad-spectrum effect on envenoming. Furthermore, we hope that this theme becomes an umbrella that considers thrombus formation, coagulation system activation, and innate and adaptive immunity as an integrated harmful process that contributes to the amplification of the pathophysiological effects triggered by the *Bothrops* venoms. Moreover, studies on the mechanisms of thromboinflammation in snakebites are crucial to better management of the victims.

## 2. Overview of the Toxic Effects of *Bothrops* Venoms

### 2.1. Local Effects

The local effects of Bothrops envenoming are considered one of the most evidenced events due to the tissue damage caused, and the peripherally limited antivenom response. Among the mechanisms involved, the inflammatory response represents a major issue that occurs during local pathophysiology [16,36,37]. The venom toxins are capable of triggering inflammation by three primary mechanisms: (i) the direct recognition of venom components (commonly known as a venom-associated molecular pattern—VAMPs) by leukocyte receptors, such as Toll-like receptors, which induce activation and mediators production; (ii) an indirect inflammatory response that is induced by damage-associated molecular patterns (DAMPs), such as cellular and extracellular matrix degradation products as a result of the venom toxin’s tissue damage; (iii) the direct activation of complement system mediators by the toxins [38,39,40].

The major clinical local manifestations are associated with inflammation cardinal signs (Figure 1). Edema is the most common inflammatory sign in *Bothrops* envenoming, which can progress to a compartment syndrome that results in disabilities to the patient [19,21]. Edema is a consequence of the leukocyte infiltration and production of mediators that signal the inflammatory response and co-option of endogenous signaling pathways, which act as positive feedback; this aggravates the inflammatory response, according to Bickler [41]. Blister formation and epithelial damage are manifestations observed in patients related to severe envenomations, as they increase the chance of infection and necrosis [42,43]. This event is caused by the action of SVMPs on extracellular matrix components and epithelial tissue, inducing the separation of skin layers (dermis–epidermis) [44,45,46]. In all cases, a rich source of damage-associated molecular patterns (DAMPs), immunomodulators, matrix metalloproteinase-9 (MMP-9), platelet degranulation, and blood coagulation were observed [42,44,47]. Pain is also commonly reported in *Bothrops* envenoming. Its presence is mainly associated with the production of neurogenic and non-neurogenic compounds at the bite site, which occurs through the direct and/or indirect action of *Bothrops* venom toxins, in addition to the activation of non-neural cells that contribute to peripheral sensitization [16]. These compounds interact with their respective receptors, altering the nociceptive signal transduction. Some of these components have already been identified, including substance P, nitric oxide, cytokines, histamine, 5-hydroxytryptamine, bradykinin, lipid-derived mediators (prostaglandins, leukotrienes, and PAF), as well as the participation of leukocyte cells produced during the inflammatory response [16,48].

Another target of the action of bothropic venom toxins includes the body’s vascular system. Arteries, veins, and other vessels of the circulatory system change in permeability and integrity, mainly due to the action of SVMPs that act on the vascular basement membrane, which consist mainly of type IV collagen, laminin, nidogen/entactin, and perlecan; this results in the formation of hemorrhagic lesions (ecchymosis and petechiae) [45,46,49,50]. Another mechanism that seems to contribute to local hemorrhage in bothropic envenoming is blood incoagulability. The venom toxins catalyze the activation of clotting factor zymogens and integrin precursors or receptors, also leading to endogenous thrombin formation, which can consequently result in disseminated intravascular coagulation followed by consumption coagulopathy [51]. However, the relationship between this phenomenon and local hemorrhage is not well understood and deserves attention. In *B. atrox* envenoming, patients with only local and not systemic bleeding showed similar levels of hemostatic factors [29].

Additionally, skeletal muscle tissue is another target of bothropic venom toxins action that iinduces muscle degeneration and myonecrosis [36,52], which can lead to temporary and chronic sequelae, and in more severe cases, amputation. However, severe rhabdomyolysis is not common in *Bothrops* envenoming as it is in Crotalus envenoming. Muscle damage is caused by the action of myotoxins, which act by altering the influx of [Ca]^2+^ ions, consequently inducing cell death [53,54]. SVMPS may also damage the local blood flow, resulting in ischemia and secondary myonecrosis [49].

### 2.2. Systemic Effects

*Bothrops* envenoming could present systemic effects, such as coagulopathy or incoagulability, platelet consumption, a hemorrhage, which lead to death if it is not promptly managed (Figure 2) [28,55,56,57,58]. Blood incoagulability is the most predominant systemic effect in *Bothrops* envenoming [28]. *Bothrops* venoms induce a high tendency to bleeding at the bite site, in the gums, and in vital organs) [28,55,56,57,58]. In addition, *Bothrops* envenoming presents thrombocytopenia, probably caused by the action of toxins in inhibiting or activating platelets, as well as forming an active surface for the coagulation cascade, which leads to increased bleeding [59].

Systemic myotoxicity is characterized by increases in serum myoglobin concentration and creatine kinase (CK) activity, hyperkalemia, and also acute kidney injury; this is mainly due to myoglobin toxicity in the renal tubules that can cause death [8,36]. The venom also causes oxidative stress, triggering an increase in lipid peroxidation, catalase, and glutathione-S-transferase activity in the liver, as well as plasma levels of aspartate aminotransferase (AST) and alanine aminotransferase (ALT), indicating hepatotoxicity [54]. In addition, pulmonary changes—respiratory failure/acute lung edema—have been reported as systemic complications associated with death in cases of *B. atrox* snakebite envenoming [17,55]. However, renal alterations are the complication that causes greatest concern in snakebites, among them acute glomerulonephritis, acute tubular necrosis, and acute renal injury (ARI). the latter is related to cases of lethality from snakebite envenoming [60]. The pathogenesis of ARI is still not fully understood. However, it is known that kidney injuries can be produced by the isolated or combined action of different ischemic and/or nephrotoxic mechanisms triggered by the biological activities of venoms in the body [8]. Studies with isolated kidneys have shown that venoms from different *Bothrops* species alter renal function parameters that vary according to the venom composition [8,61,62,63].

## 3. *Bothrops* Snakebite as a Prothrombotic State

Current evidence about the prothrombotic state of *Bothrops* envenoming is mainly based on experimental (animal studies), clinical, and laboratory data, as well as case reports [26]. *B. asper* venom has a high procoagulant potential [64]; *B. jararaca* and *B. atrox* venoms contain toxins with thrombin-like activity, which catalyze the interaction of fibrinogen with fibrin [65,66,67]. *Bothrops* venoms also have prothrombin activators; the first known being berythractivase from *B. erythromelas* venom, a non-hemorrhagic class P-III metalloproteinase [68].

Procoagulant toxins from *B. atrox* venom activate factors II, X, and V, and increase the procoagulant activity of factor VIII, which, as a result, leads to the generation of intravascular thrombin that has also been reported [69,70,71]. Another clotting factor activated by isolated components of *B. atrox* venom is FXIII [71]. In addition, the fibrinogenolytic components of *B. atrox* venom, the poison may also contribute to fibrinogen degradation and coagulopathies [72]. *B. jararaca* venom also fibrinogenolytic activity [66], whereas *B. atrox*, *B. jararaca*, and *B. neuwiedi* venoms were able to degrade fibrin [73,74,75].

Snake venom thrombin-like enzymes (SVTLEs) belong to a class of serine proteases that can cause blood clotting in vitro, a characteristic exhibited by several snake venoms. Leucurobin from *B. leucurus* venom induces the release of fibrinopeptide A and traces of fibrinopeptide B [76]; SPBA from *B. alternatus* venom and BM-IIB34 kDa + BM-IIB32 kDa from *B. moojeni* venom induce plasma coagulation, and have fibrinogenolytic activity [77], as well as pictobin from *B. pictus* and barnettobin from *B. barnetti* venom, which coagulate plasma, but also fibrinogen, releasing fibrinopeptide A [78,79]. Moojase from *B. moojeni* venom also induces clotting of platelet-poor plasma and fibrinogen solutions in a dose-dependent manner, indicating thrombin-like properties due to proteolysis of human fibrinogen Aα chains, followed by late degradation of Bβ chains [80]. 

However, some *Bothrops* venoms induce systemic multifocal thrombotic complications that are triggered by a striking feature caused by *B. caribbaeus* and *B. lanceolatus* venoms; the mechanisms involved in these vascular obstructions still need to be clarified [23,81,82]. Interestingly, thrombocytopenia, minimally reduced prothrombin with normal activated partial thromboplastin time (APTT), and elevated fibrinogen concentration, are typical findings of victims with thrombosis after a bite by *B. caribbaeus* and *B. lanceolatus*, indicating the involvement of platelets and fibrinogen with this condition [23,81,82]. The victims of snakebites by *B. atrox* with systemic hemorrhage show a reduction in the levels of factor V, II, fibrinogen, plasminogen, and alpha 2-antiplasmin in plasma, while tissue factor and FDP levels are elevated [28,29]. Consumption coagulopathy resulting from the action of *B. atrox* venom may increase the risk of systemic bleeding seen in envenomation [28]. Moreover, in *B. jararaca* envenoming, about 56% of the patients had incoagulable blood; of these, 70% had systemic bleeding without systemic bleeding, and 18% had local bleeding alone (without local bleeding) [57].

An important event that is associated with systemic thrombotic complications is the occurrence of cerebral vessel occlusion. Although it is less common than hemorrhagic stroke, ischemic stroke has been described following *B. atrox*, *B. caribbaeus*, and *B. lanceolatus* snakebites [23,81,82,83]. The evidence of neurological symptoms occurred from 24 h up to 4 days after the snakebites, and all patients received antivenom therapy; the authors suggested that delayed time to treatment or the effectiveness of antivenom could be a factor that resulted in vascular cerebral thrombus formation. These events were directly associated with venom-induced coagulation disturbances (thrombocytopenia and clotting parameters alterations). Inflammatory parameters such as leukocytes/neutrophils and C-reactive protein were also elevated in some cases. Diagnosis of the cerebral infarction evaluated by imaging exams showed that the cerebral artery (anterior, posterior, or middle segments) and basilar artery were the main arteries with thrombosis, with infarcts occurring in cortical and occipital territories. Death and permanent disabilities were observed as a possible thrombotic stroke consequence in bitten patients [23,81,82].

As a consequence of the overall alterations in hemostasis, *Bothrops* venoms are responsible for inducing thrombotic events (Figure 3). Thrombotic microangiopathy (TMA) is an important complication observed in cases of *Bothrops* accidents and preclinical studies [24,84,85,86]. The TMA is characterized by erythrocyte destruction due to small-vessel damage and microthrombi deposition, resulting in thrombotic events that can lead to acute kidney injury as the predominant end-organ damage. Clinical–laboratory evidence associates this with microangiopathic hemolytic anemia, which is characterized by red blood cell fragments (schistocytes) on the peripheral blood film, delayed thrombocytopenia, and increased levels of bilirubin and lactate dehydrogenase [87]. Although there is an association of TMA with the coagulopathy induced by experimental *Bothrops* envenomation [88,89,90], the etiology of TMA remains unclear. Studies have claimed that the TMA observed in *Bothrops* accidents is associated with hemolytic uremic syndrome, due to its tendency for renal end-organ damage; however, there is no evidence that inflammatory mediators C3, C4, or ADAM13 were altered in the patients [24,84,85].

## 4. Complement System Mediators’ Activation by *Bothrops* Venoms

The complement activity is another important pathway of the pro-inflammatory cascade that is responsible for vasodilation, chemotaxis, and leukocyte activation [91]. *Bothrops* venoms can activate the complement cascade, generating large amounts of anaphylatoxins, such as C3a, C4a, and C5a, which are considered to be the bridge between innate and adaptive immunity. Alternative pathway activity is completely inhibited by the venoms of *B. atrox*, *B. cotiara*, *B. moojeni*, *B. itapeningae*, *B. paradoi*, *B. hyoprorus*, *B. insularis*, *B. bilienatus*, *B. brazili*, *B. jararaca*, and *B. marajoensis*. However, this effect is not observed for the venoms of *B. fonsecai*, *B. taeniata*, *B. alternatus*, *B. leucurus*, *B. erythromelas*, *B. jararacussu*, and *B. neuwiedi*, which do not affect the alternative pathway (Figure 4) [92].

Delafonatine et al. [93] found that the *B. lanceolatus* venom, a Martinique native species, potentially activates the complement system and dose-dependently reduces the lytic activity of the alternative pathway. *Bothrops* venom also activates the classical complement pathway by cleaving the C1 inhibitor by proteases of this venom, interrupting the control of complement system activation [92]. The classical, lectin, and alternative pathways converge when an additional C3b protein associates with C2bC4b or C3bBb, which creates a C5 convertase that converts C5 to C5a and C5b. *B. lanceolatus* venom induces a significant production of C5a that is capable of inducing an influx of calcium in leukocytes. However, inhibition occurred in the presence of 1,10-phenanthroline, suggesting that C5 cleavage and consequent C5a release occur by the action of metalloproteases present in the venom [93]. C5b recruits C6 followed by C7, C8, and various C9 proteins to insert into the target cell membrane to form the MAC complex, also called the terminal complement complex, a pore in the membrane to induce cell lysis [94,95]. In addition to the formation of the MAC complex, the complement system has other functionalities, including opsonization, NETose, and the production of the so-called anaphylatoxins (C3a and C5a) [94,96,97].

## 5. Role of Neutrophils

Neutrophils are the first line of defense of the innate immune system [98], of which the ability of *Bothrops* venoms to stimulate these cells is a striking feature of great relevance in the context of inflammation. Neutrophils are present in myonecrotic and hemorrhagic areas, or even in the inflammatory infiltrate [31]; once activated, they produce pro-inflammatory cytokines, phagocytose, and release extracellular neutrophil traps (NETs), acting on the repair process, or potentially causing tissue damage [99,100,101,102,103,104,105]. The effects of *Bothrops* venoms on neutrophils have been explored, generating a large volume of information over the decades [31]. In vivo studies have shown that in inflammatory events, venoms can induce the migration of polymorphonuclear neutrophils to the envenoming sites of *B. alternatus* [106,107], *B. asper* [108,109], *B. atrox* [110], *B. erythromelas* [106], *B. jararaca* [109,111], *B. jararacussu* [112], *B. lanceolatus* [113], *B. taeniata*, and *B. bilineata* [114].

Neutrophilia and high concentrations of NETs in circulating blood, and thrombocytopenia, are described as an indicator of severity and a poor clinical outcome in thromboinflammatory diseases [115,116]; thus, they may be indicators for thromboinflammation and predictors of a poor clinical outcome during envenoming. It is known that patients with severe tissue damage caused by *B. atrox* venom have an immunological profile that is polarized to the Th1 response, and present a more intense local immune response with high levels of IL-1β, IL-6, TNF-α, MIP-1 (CXCL-1), and MIP-2 (CXCL-2) [43,110]. However, the Th1 response caused by *B. atrox* venom is regulated by neutrophils and the myeloid differentiation factor 88 (MyD88) pathway [40]. In addition, neutrophilia and thrombocytopenia of varying intensities have been reported in *Bothrops* snakebite envenoming, but more intense changes have been reported in cases that had tissue loss and/or limb amputations [19,21,22].

## 6. Cytokines and Their Role in Inflammation: An Overview

The cytokine storm is a systemic inflammatory syndrome characterized by the emergence of multiple disorders in the regulation of the immune response [117,118,119,120]. During the NETosis process, neutrophils can amplify the production of cytokines, while activated macrophages secrete large amounts of cytokines, which can cause organ damage. On the other hand, NK cells have their cytolytic function attenuated during the cytokine storm, hindering the process of inflammation [121].

Although diverse, only some subtypes of T cells are also implicated in and/or influenced by the cytokine storm. Thus, the ability of cytotoxic T lymphocytes (CTLs) to kill damaged and/or infected cells is impaired, which causes prolonged T cell activation that culminates in cascades of tissue damage at the site of inflammation [122]. A complex and interconnected network of cell types, signaling pathways, and cytokines is involved in cytokine storm disorders. Important crosstalk refers to the participation of plasma proteins, such as complement proteins and other inflammatory mediators, which contribute to the genesis and amplification of cellular responses, and provide feedback on cytokine signaling. In this way, cytokines can induce an increase in the production of complement proteins, which in turn can increase or inhibit the production of cytokines, which can cause collateral damage if excessive.

The inflammatory process caused by *Bothrops* venoms involves leukocytes infiltration, mainly polymorphonuclear and/or mononuclear cells at the site of injury, and the involvement of other resident cells that produce and release cytokines in response to *Bothrops* snake venom (Table 1) [123]. *Bothrops* venom toxins are also capable of inducing cytokine synthesis. Cytokine release in the kidney increased by the action of Asp-49 PLA_2_ (IL-10) and Lys-49 PLA_2_ (TNF-α, IL-1β, IL-10) from *B. pauloensis* venom [62], while levels of TNF-α, IL-1β, and IL-10 increased in isolated kidneys perfused with *B. alternatus* venom [124]. BJ-PLA_2_-I, a PLA_2_ Asp49 from *B. jararaca* venom, was found to induce increases in IL-6 and IL-1β in the inflammatory exudate of mice [125]. The injection of myotoxic PLA_2_s (Lys-49 and Asp-49) and *B. asper* venom SVMP also promoted increases in the concentrations of IL-1, IL-6, and TNF-α in the peritoneal exudate of mice [126,127]; however, intramuscularly, these toxins induced increases in IL-6 and IL-1β in muscle tissue, but not in TNF-α [118].

Furthermore, soluble levels of chemokines (CXCL-8, CCL-5, CXCL-10, and CCL-2) and cytokines (IL-6, IL-1,0, and IL-2) are higher in victims of *B. atrox* bites, with some tissue complication, and can more significant based on the severity of the damage. In addition, patients with more severe complications were found to present a profile of individuals with a high production of the molecules CXCL-9, IL-6, and IL-10, while the group with severe complications presented the profile of a high production of molecules CXCL-8, CXCL-9, CCL-2, CXCL-10, IL-6, IL-1β, IL-10 and IL-2, a profile that is opposite to that observed in healthy individuals [43]. The *Bothrops* snakebite inflammatory response was shown to be responsible for modulating early adverse reactions (EARs) associated with antivenom therapy. Soares and colleagues [133] observed that patients who presented increased levels of CXCL-8 and IL-2 were less susceptible to developing EARs, suggesting the involvement of both chemokines/cytokines during the onset of antivenom adverse reactions. Patients with a cytokine storm have high-degree fevers in severe cases. Fever can be induced by interleukin-1, interleukin-6, or TNF, through different mechanisms; more severe cases can develop kidney failure, liver damage, and other outcomes (Figure 5). The specific markers are restricted to the dosage of circulating cytokines, and the laboratory findings from cytokine storms are variable and influenced by the cause, although nonspecific markers of inflammation, such as C-reactive protein (CRP), show an increase that is correlated with severity [122,134,135].

## 7. Platelets and *Bothrops* Venoms

Platelets are enucleated cells that are essential for hemostasis and predominant cellular elements in the thromboinflammation process [136]. After an injury, platelets perform a wide variety of functions, which include adhesion to endothelial and subendothelial structures, followed by activation and aggregation, constituting the initial hemostasis response [137]. Once platelets become aggregated, the coagulation cascade is triggered by either intrinsic or extrinsic pathways, leading to the activation of prothrombin to thrombin which converts fibrinogen molecules to fibrin [137]. Thrombin is considered to be a central mediator of thromboinflammation; it activates platelets by cleaving protease-activated receptors (PARs), leading to thrombus formation [138]. This mediator affects the components of the vasculature through the cleavage of components of the coagulation, complement, and fibrinolytic systems, as well as the activation of endothelial cells, leukocyte migration, macrophage activation, expression of vascular endothelial growth factor in smooth muscle cells and others (reviewed in [139]). In addition to PARS, a wide variety of receptors are found on the platelet surface (classified as integrins, selectins, and receptors of the immunoglobulin type) that allow platelet multifunctionality in hemostasis, inflammation, thromboinflammation, and resolution [140,141,142,143]. The engagement of platelet receptors allows platelets to act as contact elements between complement, coagulation, and contact systems [144]. In addition, these functionally multifaceted cells also have the ability to establish interactions with leukocytes in inflammation [136,142].

Platelet–complement interaction is mediated by several complement factors, including a form of C3 and other complement receptors such as C3aR and C5aR, and complement binding evokes an activation response in platelets. When activated, platelets can also activate the complement cascade, resulting in its amplification [142,145]. In addition, platelets, through the lectin pathway, affect the coagulation system by activating MASP-1, which can exert thrombin-like activity [146,147]. However, this interaction represents highly complex and multifactorial crosstalk, which is beyond the role of C3, C3aR, and C5aR. To illustrate, P-selectin is a platelet activator of the alternative complement pathway [148], of receptors binding to complement components such as C1q [149], the activation of C4 [150], the deposition of C3b and C5b-9 on activated platelets, of complement activation by platelets without thrombin production, and reduced platelet C3 activation in the absence of C1 or Factor B [150], representing other examples by which platelets can trigger the complement system.

Platelet–leukocyte interaction is mediated by P-selectin expressed on the surface of activated platelets, which can interact with the PSGL-1 selectin receptor expressed on leukocytes. At the site of inflammation, platelets aid leukocyte emigration by capturing leukocytes at specific extravasation sites, thus facilitating tissue infiltration in a PSGL-1/P-selectin-dependent manner [151]. Neutrophils actively search for activated platelets to engage in a PSGL-1-mediated signaling event [152]. In addition, activated platelets express the integrin GpIIb/IIIa, enabling binding to fibrinogen, leading to crosslinking with neutrophils mediated by their surface integrin receptor Mac-1 that can directly interact with the platelet receptor GpIbα, and thus regulating thrombosis [153]. Other platelet–leukocyte interactions are also known to modulate prothrombotic and proinflammatory pathways [142]. 

The role of platelets in inflammatory responses modulated by *Bothrops* venoms has been reported [30,154]. Despite toxins from *Bothrops* venom that exist with the ability to disrupt platelet activation, the entire ability of the venom to induce this process has been seldom explored. *B. jararaca* venom activates platelets, causing platelet sequestration and pulmonary thrombin development composed of platelet aggregates and dense fibrin sheaths, in addition to fibrin in the kidneys. Within this framework, it is important to highlight that the combination of pro-inflammatory and thrombotic scenarios can lead to organ failure due to excessive platelet activation, coagulation, and fibrin deposition in the microvasculature [155]. PLA_2_ isolated from *Bothrops* venoms has already been cataloged with the potential to inhibit platelet aggregation. BmooTX-I [156] and BmooPLA2 [157] isolated from *B. moojeni* venom, BthA-I-PLA_2_ from *B*. *jararacussu* [158], BJ-PLA_2_ from *B. jararaca* [159], BE-I-PLA_2_ from *B. erythromelas* [68], and BpPLA_2_-TXI of *B. pauloensis* [160] represent some examples. However, also noteworthy is the ability of bothrotropstoxin-II (BthTX-II), an Asp49 PLA_2_ isolated from *B. jararacussu* venom [161], to induce platelet aggregation. When approaching *Bothrops* SVMPs, an interesting case to present is jararhagin. Jararhagin is a P-III SVMPS from B. jararaca venom that inhibits platelet aggregation induced by ristocetin and collagen [56]. In addition, jararhagin dramatically reduces α2β1 integrin on the surface of platelets [162]. Alternagin from *B. alternatus* [163] also inhibits collagen-induced platelet aggregation, and barnettlysin-I from *B. barnetti* venom causes platelet inhibition induced by ristocetin and collagen [164]. In addition to PLA_2_ and SVMPs, other toxins from *Bothrops* venoms have already been isolated, and their roles on platelets have already been explored (Table 2).

The platelet counts of patients in less severe cases appear to rapidly normalize after administration of the antivenom. The increase in platelet count after antivenom treatment may be related to the production of interleukin-6, a potent promoter of thrombopoiesis and megakaryocytopoiesis [27,185,186,187]. This has already been reported in experimental models using the venom of *B. asper*, *B. jararaca* [188], and in human envenomations by *Bothrops* from Costa Rica and Brazil [189,190].

Victims of *B. atrox*, before antivenom therapy, have a platelet count that correlates poorly with tissue factor, factor II, factor V, D-dimer, plasminogen activity, and a moderate correlation with fibrinogen and FDP. In addition, platelet counts are lower in patients with systemic bleeding [28,29]. Thus, the combination of thrombocytopenia and changes in clotting factor levels increases the risk of bleeding, suggesting crosstalk between these systems. Thrombocytopenia of varying intensity is found in victims of *B. jararaca* bites, but whether or not thrombocytopenia is associated with spontaneous systemic bleeding depends on the venom involved [27]. Victims of *B. atrox*, prior to antivenom therapy, have a platelet count that correlates poorly with tissue factor, factor II, factor V, D-dimer, plasminogen activity, and a moderate correlation with fibrinogen and FDP. In addition, platelet counts are lower in patients with systemic bleeding [29]. Thus, the combination of thrombocytopenia and changes in clotting factor levels increases the risk of bleeding, suggesting crosstalk between these systems.

## 8. Snake Venom-Induced Coagulation/Inflammation Cross-Talk and Participation of Tissue Factor

It is well known that both inflammation and the hemostatic system are highly integrated, and the unbalance of these events is responsible for several pathological conditions. An inflammatory response can trigger coagulation (and vice versa) [191]. The independent mechanisms of the inflammatory response and coagulation disturbance induced by *Bothrops* venoms are well known; however, their association still is a growing issue in the toxinology field. A recent study on *B*. *atrox* snakebite patients from the Brazilian Amazon has evaluated the inflammation/coagulation cross-talk [192]. By comparing inflammatory mediators’ profiles and fibrinogen consumption, the authors reported that the cytokines/chemokines CXCL-8, CXCL-9, CCL-2, and IL-6 are directly affected by fibrinogen levels, and increased levels of CCL-5 and decreased levels of IFN-γ were observed in patients with hypofibrinogenemia. The study was the first report on the inflammatory response and coagulation mutual relationship involving *B*. *atrox* snakebite patients, leading the way for other studies [192].

The interface of the immune response and hemostasis consists of a wide range of reactions characterized by the involvement of cellular events and blood mediators. Among the molecules, tissue factor (coagulation factor III) is considered to be a hallmark in the inflammation/coagulation cross-talk. Tissue factor (TF) is a transmembrane protein constitutively expressed in perivascular cells where it forms a hemostatic barrier. However, upon inflammatory stimulus, TF is expressed mostly by endothelial cells and monocytes within the intravascular compartment, and interacts with circulating factor VII to generate the extrinsic tenase complex, thus activating FX and triggering blood clotting [193,194]. Considering snake venoms, pre-clinical and clinical studies have shown the involvement of *Bothrops* venoms and toxins are capable to induce the expression of TF. The only clinical study showing the involvement of TF was conducted with patients following *B*. *atrox* snakebites [29]. The study showed an increase in plasma levels of TF antigen, which presented correlations with hemostatic components such as coagulation factors, platelets, and the fibrinolysis system. The authors also demonstrated that patients with systemic bleeding presented higher levels of TF compared to those without bleeding, and that TF levels also significantly increased in patients with moderate/severe edema when compared to those with mild edema. These data suggest the expression of intravascular TF, as well as extravascular TF released by local trauma and damage caused by envenomation [29]. Similar findings were also observed by Yamashita and colleagues [195], using an experimental approach, showing that both local (subcutaneous) and systemic (intravenous) administration of *B. jararaca* venom in mice increased plasma TF activity, which was confirmed by increased skin (local) and lung (systemic) TF expression [195]. 

In mechanisms of venom-induced TF participation, SVMPs have been found to play an important role. Yamashita and colleagues [195] observed that SVMPs are the major class of toxins within *B. jararaca* to induce TF activity in mice. In vitro, berythractivase and moojenactivase, both PIII SVMPs isolated from *B. erythromelas* and *B. moojeni*, respectively, were capable to induce TF expression on vascular cells. Berythractivase, a prothrombin activator, induced TF activity and gene expression in endothelial cells [196]. Moojenactivase, a prothrombin and factor X activator, was capable of inducing TF expression of peripheral blood mononuclear cells (PBMC), in both in vivo and in vitro, which led these cells to adopt pro-coagulant behavior [89,176]. Concerning the inflammation/coagulation interface, berythractivase was also found to up-regulate endothelial ICAM-1 gene expression, as well as nitric oxide (NO) generation, prostaglandin I2 (PGI2) and interleukin-8 (IL-8) release [197]. As for moojenactivase, the toxin was capable of inducing TF production/activity along with inflammatory mediators TNF-α, CXCL-8, and CCL-2 in PBMC in vitro, and leukocyte, IL-6, and TNF-α in mice [89,176]. Aside from SVMPs, Cezarette and colleagues [197] demonstrated that BjcuL, a galactoside-binding lectin, induced TF production/activity mediated by its pro-inflammatory activity. BjcuL was capable of interacting and activating Toll-like receptor 4 (TLR4) to induce IL-1β, TNF- α, and IL-6 expression, as well as TF in monocytes in vitro. When the cells were preincubated with galactose, TLR4, and NFκB antagonist/inhibitor, not only were inflammatory mediators mitigated, but also TF [198].

Considering the studies described above, the inflammatory response induced by *Bothrops* venoms is also responsible for triggering procoagulant behavior. Aside from the direct activation of hemostatic events by venom toxins, the inflammation acts as a potentiating factor to improve coagulation disturbances during snakebites.

## 9. Thromboinflammation and Snakebite

Thromboinflammation has been used to refer to the activation of the cascade systems present in the blood, as well as the activation of its multicellular system [32]. However, thromboinflammatory pathways can exacerbate inflammation and cellular interactions, which can lead to vaso-occlusion, ischemia/reperfusion, and eventually irreversible organ damage [199]. The cascade and multicellular blood systems are involved in the immediate response to snake venoms, and although classically seen as independent systems, their interrelationship has been associated with the amplification of venom toxicity [33,34,35]. The characterization of *Bothrops* envenoming as a prothrombotic, inflammatory state with multiple blood cell activation is supported by extensive experimental, clinical, and laboratory data, as well as case reports. From an experimental point of view, several components of the hemostatic system have already been characterized as direct and/or indirect targets of *Bothrops* venoms and/or their toxins. The same can be said for the complement system, whose different venoms can cleave key effectors that cause complement activation via the classical, alternative, or lectin pathways. Although there are some inconsistencies between the results, possibly reflecting differences in the composition of the venoms or, for example, issues related to the method, processing, and analysis of samples, it can be safely stated that *Bothrops* envenoming is a condition in which the hemostasis-complement-blood cells are affected.

However, the complex interaction between hemostasis and activation of innate immunity makes it difficult to precisely define the relative contribution of each of these two processes to the pathogenesis of different complications, possibly explaining the absence of a direct association between the classic biomarkers of hemostasis activation and the risk or severity of some of these clinical manifestations [200]. Therefore, the crossing of inflammation and thrombosis is very well exemplified during snakebite envenoming, due to the presence of a wide variety of characterized proteins that can activate the innate immune system and/or hemostasis. Evidence supports the cross-referencing of inflammation with hemostasis: (i) studies in animal models report disturbances of hemostasis; despite some heterogeneity within the model and within the venom, a less effective hemostatic system is associated with an increase in hemorrhagic manifestations; (ii) it has been demonstrated that components of the coagulation system, such as platelets, also signal via immunological pathways; (iii) there are examples of toxins and venoms whose mechanisms disturb the local hemostatic balance and induce inflammation; and (iv) studies show that leukocytes are not only found at the site of envenomation, but also in arterial and venous thrombi.

Complement activation and primary hemostasis can be linked to the moment of initiation, as a direct interaction of vWF and C1q was recently described [201]. The formation of platelet–leukocyte aggregates is highly relevant to cardiopulmonary bypass, which is mediated by C5a [202,203]. The formation of platelet–leukocyte aggregates, which is considered an important event in several contexts of thromboinflammation, is mediated by the alternative complement pathway and, specifically, by its regulator properdin; this suggests that the complement system may play a role in thrombocytopenia [204,205]. The reproduction of this phenomenon by the venom of other species, the mechanisms by which *Bothrops* venoms induce thrombocytopenia, and the possible relationship of the complement system on platelets, need elucidation. Focusing on the pathophysiological and translational relevance of a complement–platelet interaction hypothesis, it is noteworthy that complement activation is relevant in several inflammatory conditions associated with injury. Moreover, although platelets have been suggested as a key mediator of thromboinflammation in envenomation, complement activation can also be considered of equal relevance, mainly because of the evidence that has shown platelet activation by the complement system, as well as the ability of the complement pathway to be modulated by platelets.

## 10. Conclusions

The inflammatory response and coagulation disorders are considered hallmark mechanisms associated with local and systemic effects during *Bothrops* snakebite. However, the association of both events still is a recent issue in toxinology, although this is well established in other diseases. The crosstalk is responsible for potentiating both inflammatory and hemostatic alterations, enhancing prothrombotic conditions associated with thrombotic microangiopathy and tissue ischemia. Therefore, studies on the crosstalk between hemostasis/inflammation and thromboinflammatory events should be encouraged, seeking to highlight new mechanisms to understand their role in the pathophysiology of envenomation. The search for thromboinflammatory markers with predictive or prognostic roles can provide support to improve the clinical condition. Additionally, the progress made in better understanding the coagulation/inflammation interface enlightens the prospects to novel therapeutical approaches associated with antivenom therapy. Finally, we believe that this new aspect may elucidate previously unexplored paths in the complex pathology triggered by snake venoms and toxins, and may allow the discovery of new therapeutic targets and procedures to address the mortality and morbidity of envenomation.

## Figures and Tables

**Figure 1 ijms-24-11508-f001:**
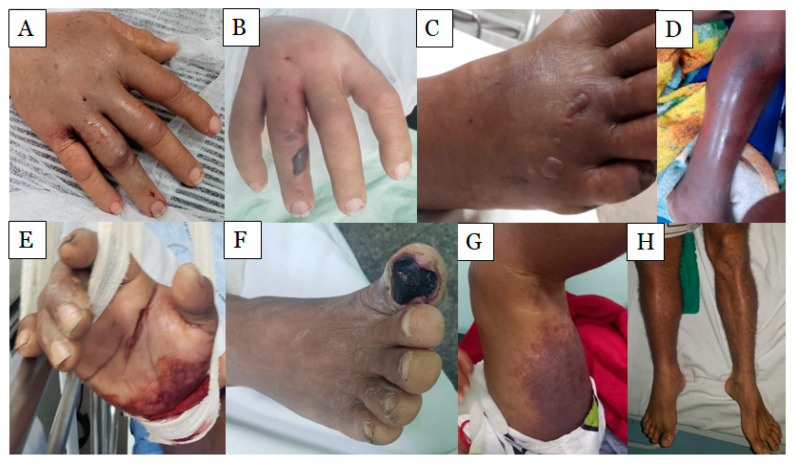
Local clinical manifestations of *Bothrops atrox* snakebite envenoming. (**A**) Edema on the right hand; (**B**) blisters on the right hand; (**C**) blisters on the right foot; (**D**) edema on the right leg; (**E**) right wrist bleeding (bite site in the joint of the hand with the arm); (**F**) necrosis in the right hallux; (**G**): ecchymosis on the entire left thigh, distant from the bite site; (**H**): right limb swelling. Photos were taken by Lisele Brasileiro.

**Figure 2 ijms-24-11508-f002:**
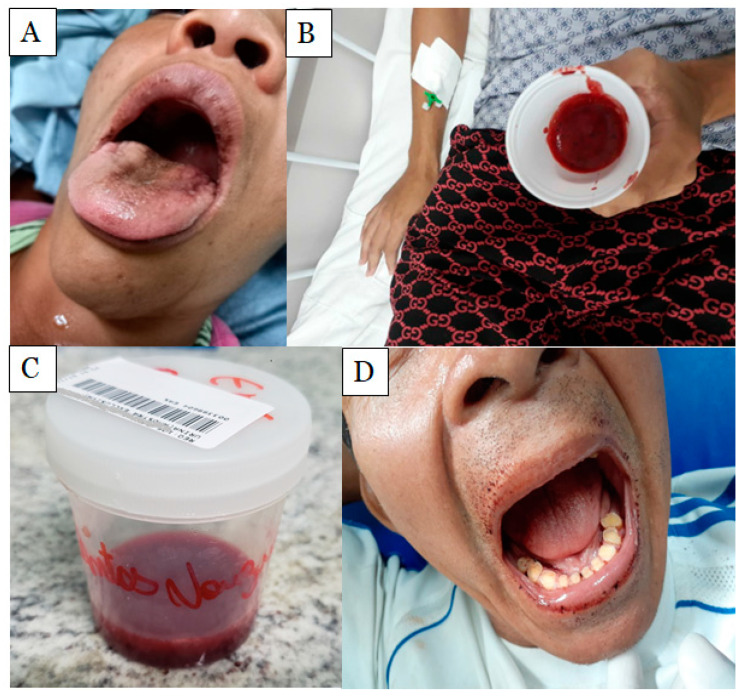
Clinical systemic manifestations of *Bothrops atrox* snakebite envenoming. (**A**) Patient after hematemesis; (**B**) hemoptysis in a male patient; (**C**) macroscopic hematuria; (**D**) gum bleeding. Photos were taken by Lisele Brasileiro.

**Figure 3 ijms-24-11508-f003:**
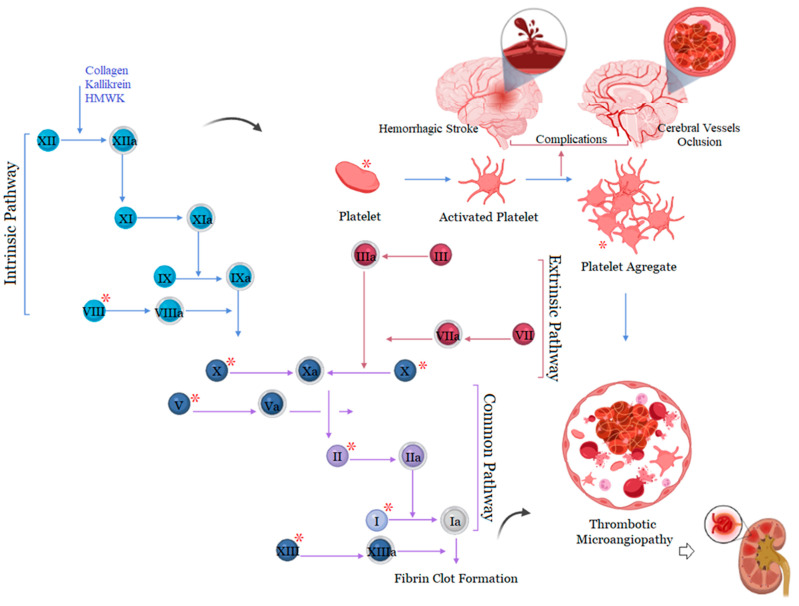
Components of the hemostatic system targeted by toxins and several consequences of general changes in hemostasis during *Bothrops* snakebite envenoming. * Components of the hemostatic system that are affected by *Bothrops* toxins. Created with BioRender.com, accessed on 30 September 2022 by Joeliton S. Cavalcante.

**Figure 4 ijms-24-11508-f004:**
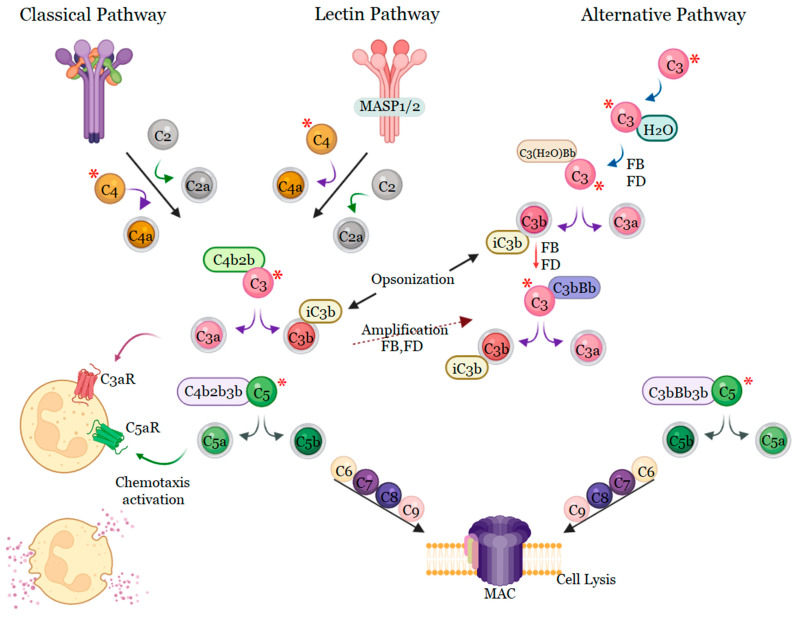
Mechanisms of activation of the complement system by *Bothrops* venoms. *Bothrops* venoms activate the complement system via classical pathways, some also via lectins or the alternative pathway. All *Bothrops* venoms cleave C3 and C4, resulting in the synthesis of anaphylotoxins C3a, C4a, and C5a, and the terminal complement complex causing activation of chemotaxis. * Components of the complement system that are affected by *Bothrops* venoms. Created with BioRender.com, accessed on 30 September 2022 by Joeliton S. Cavalcante.

**Figure 5 ijms-24-11508-f005:**
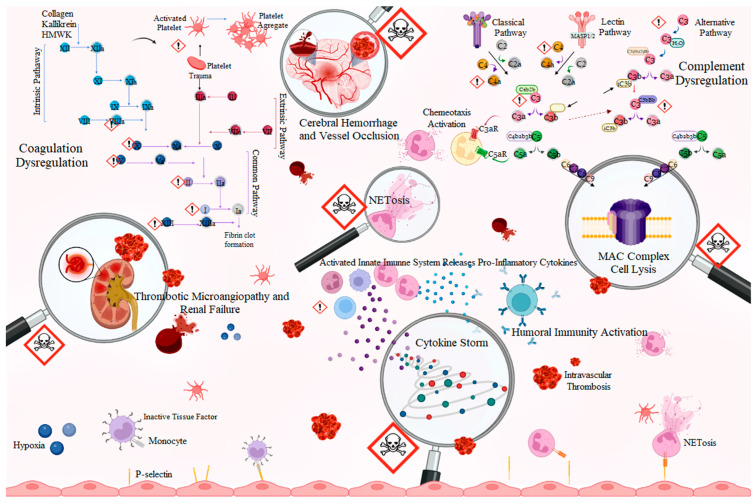
Thromboinflammatory pathways in *Bothrops* envenomation. The direct action of toxins on components of the hemostatic and complement system leads to hemostatic and inflammatory changes in *Bothrops* envenomation, which can result in clinical complications responsible for cases of tissue loss and death. *Bothrops* venoms act directly on clotting factors, causing the activation of the cascade that culminates in the formation of fibrin. During fibrin formation, clotting factors are consumed, resulting in disseminated intravascular coagulation. The generated fibrin induces the formation of intravascular thrombin, and plays a critical role in thrombotic microangiopathy through the lysis of erythrocytes that collide with it, which can lead to acute renal failure. The activation of the complement system modulated by the cleavage of key factors by the action of venoms amplifies the toxicity of the venom through chemotaxis and the activation of leukocytes, especially neutrophils. Neutrophils can form NETs that damage local tissue which has already been affected by the venom entering the body. In addition, the activation of the complement system culminates in the formation of the membrane attack complex that potentiates local tissue damage. Endothelial activation by platelets, leukocytes, and ischemia–reperfusion events result in the expression of adhesion molecules, including P-selectin, which recruits leukocytes and, in turn, red blood cells to blood vessel walls. The recruitment of leukocytes, platelets, and red blood cells to the vascular wall, together with the clotting processes, extracellular neutrophil trap components, and the formation of heterocellular aggregates between platelets, leukocytes, and red blood cells, with the subsequent entrapment of red blood cells; this results in the cerebral vaso-occlusive processes. In other cases, due to thrombocytopenia, cerebral hemorrhage occurs. Exclamation points represent targets of toxins. Skulls represent clinical complications associated with the effects of venom. Loupes indicate the need to study the consequences of clinical complications that are poorly understood. Created with BioRender.com, accessed on 4 June 2023 by Joeliton S. Cavalcante.

**Table 1 ijms-24-11508-t001:** Some examples of the effects of *Bothrops* snake venoms in the synthesis and release of cytokines in different experimental models.

Venom	Model	Cytokines	Reference
*B. alternatus*	mice peritoneal macrophages	TNFα, IL1, IL12 and IL6	[107]
*B. alternatus*	Raw 264.7 cells	TNFα, IL1, IL12 and IL6	[107]
*B. asper*	peritoneal cavity of mice	IL-6 and TNF-a	[126]
BaP1 *from B. asper venom*	macrophages in vitro	TNF-a	[128]
*B. atrox*	peritoneal cavity of mice	TNF-a, IL-6, IL-12p70, IL-10 and CCL-2	[129]
*B. erythromelas*	mice splenocytes	IFN-γ, IL-6, IL-10 and NO	[130]
*B. jararacussu*	footpad samples	TNF-α and IL-1β	[112]
*B. lanceolatus*	human keratinocytes—HaCaT	IL-8, MCP-1, RANTES	[131]
*B. lanceolatus*	endothelial vascular cells—EAhy926	IL-8, MCP-1, RANTES e IL-6	[131]
*B. moojeni*	spinal cord	IL-10	[132]
*B. moojeni*	footpad samples	IL-6, IL-10 and TNF-α	[132]

**Table 2 ijms-24-11508-t002:** Effects of toxins in *Bothrops* snake venoms on platelets.

Protein Family	Protein Name	Species	Inhibition (−)/Activation (+) of Aggregation	References
PLA_2_	Bothropstoxin-II	*B. jararacussu*	+	[161]
	BmooPLA_2_	*B. moojeni*	−	[157]
	BJ-PLA_2_	*B. jararaca*	−	[136]
	BthA-I- PLA_2_	*B. jararacussu*	−	[158]
	BE-I-PLA_2_	*B. erythromelas*	−	[68]
	BpPLA_2_-TXI	*B. pauloensis*	−	[160]
	BmooTX-I	*B. moojeni*	−	[156]
	Braziliase-I and II	*B. brazili*	−	[165]
	BJ-PLA_2_-I	*B. jararaca*	−	[125]
	BaltPLA_2_	*B. alternatus*	−	[166]
SVSP	BpirSP27	*B. pirajai*	+	[167]
	BpirSP41 B	*B. pirajai*	+	[167]
	TLBm	*B. marajoensis*	+	[168]
	PA-BJ	*B. jararaca*	+	[169]
	Thrombocytin	*B. jararaca*	+	[169]
	Cerastotin	*B. jararaca*	+	[170]
	Bothrombin	*B. jararaca*	+	[171]
	Moojase	*B. moojeni*	+	[80]
SVMP	BmooMPα-II	*B. moojeni*	−	[172]
	Jararhagina	*B. jararaca*	−	[173]
	BmooPAi	*B. moojeni*	−	[174]
	Bar-I	*B. barnetti*	−	[164]
	Batroxrhagin	*B. atrox*	−	[74]
	BaltDC	*B. alternatus*	−	[175]
	Moojenactivase	*B. moojeni*	+	[176]
	Atroxlysin-III	*B. atrox*	−	[177]
	r-colombistatins 2,3,4	*B. colombiensis*	−	[178]
LAAOs	BpirLAAO-I	*B. pirajai*	+	[179]
	Bp-LAAO	*B. pauloensis*	+	[180]
	BmooLAAO-I	*B. moojeni*	+	[181]
	Bl-LAAO	*B. leucurus*	−	[182]
CTL	Botrocetin	*B. jararaca*	+	[183]
	Baltetin	*B. alternatus*	−	[184]

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
