# Peer review of "Crosstalk of Inflammation and Coagulation in *Bothrops* Snakebite Envenoming: Endogenous Signaling Pathways and Pathophysiology"

_ijms, 2023, doi:10.3390/ijms241411508_

Round 1

Reviewer 1 Report

The review by Cavalcante et al described pathophysiological effect of Bothrops envenoming on coagulation, complement, platelets, neutrophils, and inflammation of endothelium. The review is detailed and is highly relevant to the readers due to the prevalence of snakebites caused by the Bothrops.

I have only a few minor comments, hopefully these will help the authors in their revision of the manuscript:

1. Line 151-152: This sentence is difficult to understand, what are the two similar conditions being compared here? Are the authors comparing patients with local vs without systemic bleeding? But local bleeding = without systemic bleeding, so if ‘local and without systemic bleeding’ is one group, then this sentence is incomplete.

2. Line 190: What is the patient consent for with respect to this manuscript? I assumed it’s for the photography. If that is the case, is they an approval number from the relevant ethics board? And this should be a standalone section instead of 2.1 (under ‘Overview of the toxic effects of Bothrops venoms’).

3. Line 205-206: Incomplete sentence.

4. Line 240: “venom-induced coagulopathy, evidenced…”

5. Line 241: “markers such as leukocytes/neutrophils…”

6. Fig 3: maybe also put an asterisk on the 2nd factor X (below platelet) and the arrow that is perpendicular to the arrow originated from factor Va should originate from Xa. Also maybe consider putting an asterisk on platelet aggregation to indicate many Bothrops toxins directly modulate aggregations.

7. Fig 4: there are toxins directly cleaves C5 right? Then maybe putting the asterisk on C5 too.

8. In many places throughout the manuscript, the ‘2’ in PLA2 should be a subscript. 

9. The section on platelets should preferably be moved to be together with bleeding/thrombosis to group haemostatic effects. 

10. Fig. 5: cell types should be more clearly labeled. 

Please see above.

Author Response

# Reviewer 1

Comments and Suggestions for Authors

The review by Cavalcante et al described pathophysiological effect of Bothrops envenoming on coagulation, complement, platelets, neutrophils, and inflammation of endothelium. The review is detailed and is highly relevant to the readers due to the prevalence of snakebites caused by the Bothrops.

I have only a few minor comments, hopefully these will help the authors in their revision of the manuscript:

  1. Line 151-152: This sentence is difficult to understand, what are the two similar conditions being compared here? Are the authors comparing patients with local vs without systemic bleeding? But local bleeding = without systemic bleeding, so if ‘local and without systemic bleeding’ is one group, then this sentence is incomplete.

Response: The sentence was rewritten.

  1. Line 190: What is the patient consent for with respect to this manuscript? I assumed it’s for the photography. If that is the case, is they an approval number from the relevant ethics board? And this should be a standalone section instead of 2.1 (under ‘Overview of the toxic effects of Bothrops venoms’).

Response: Information related to ethical aspects has been moved to an appropriate section (line 623 - 629).

  1. Line 205-206: Incomplete sentence.

Response: The sentence was rewritten.

  1. Line 240: “venom-induced coagulopathy, evidenced…”

Response: The sentence was rewritten.

  1. Line 241: “markers such as leukocytes/neutrophils…”

Response: The sentence was rewritten.

  1. Fig 3: maybe also put an asterisk on the 2nd factor X (below platelet) and the arrow that is perpendicular to the arrow originated from factor Va should originate from Xa. Also maybe consider putting an asterisk on platelet aggregation to indicate many Bothrops toxins directly modulate aggregations.

Response: Thanks for the suggestion. The figure has been modified.

  1. Fig 4: there are toxins directly cleaves C5 right? Then maybe putting the asterisk on C5 too.

Response: Thanks for the suggestion. The figure has been modified.

  1. In many places throughout the manuscript, the ‘2’ in PLA2 should be a subscript.

Response: The terms was rewritten, and all ‘2’ in PLA2 should be a subscript.

  1. The section on platelets should preferably be moved to be together with bleeding/thrombosis to group haemostatic effects.

Response: Thanks for the suggestion. However, because the section presents the role of platelets in poisoning beyond hemostasis, we prefer to keep the section on the role of platelets in poisoning. The section also highlights the inflammatory effects (effects on the complement system and leukocytes) during poisoning, and considering platelets as a key element of cross-talk inflammation-coagulation, a section addressing this is worthy.

  1. Fig. 5: cell types should be more clearly labeled.

Response: We would like to make this change. However, due to the size of the original figure, and its richness of detail, reducing it would compromise the quality of the elements.

Reviewer 2 Report

Strengths:

The article provides a comprehensive review of the current understanding of the pathophysiology of Bothrops snakebite, particularly focusing on the interaction between hemostatic and inflammatory mechanisms.

The article incorporates a wide range of sources, including both preclinical and clinical studies, to support its conclusions.

The article proposes potential areas for future research, such as the search for thromboinflammatory markers with predictive or prognostic roles.

The article includes helpful figures and diagrams to aid in understanding the complex interactions between various components of the hemostatic and complement systems.

Weaknesses:

The article's language is technical and may be difficult for non-experts to understand.

While the article provides a comprehensive review of the literature, it does not include any original research or novel findings.

The article does not discuss the limitations of the studies it cites or address potential confounding factors that may affect the results.

The article does not explore potential differences in the pathophysiology of Bothrops snakebite among different populations or geographic regions.

The article's focus on Bothrops snakebite may limit its applicability to other types of snakebite envenomation.

Questions for the authors:

Have you identified any potential areas of research that you believe could lead to improved treatments for Bothrops snakebite envenomation?

How does the pathophysiology of Bothrops snakebite compare to that of other types of snakebite envenomation?

Have you observed any differences in the pathophysiology of Bothrops snakebite among different populations or geographic regions?

Are there any potential confounding factors that may affect the results of the studies you cite in the article?

How can your findings be translated into clinical practice to improve outcomes for patients with Bothrops snakebite envenomation?

Suggestions for improvement:

The authors could consider using more accessible language to make the article more approachable for non-experts.

The authors could include a discussion of potential limitations or confounding factors that may affect the interpretation of the studies they cite.

The authors could explore potential differences in the pathophysiology of Bothrops snakebite among different populations or geographic regions.

The authors could consider incorporating original research or novel findings to complement their literature review.

The authors could broaden their focus beyond Bothrops snakebite to explore the pathophysiology of other types of snakebite envenomation.

Overall, I believe that this article is worthy of publication in a peer-reviewed journal, but it could benefit from some revisions and additional analysis.

Strengths:

 The article provides a comprehensive review of the current understanding of the pathophysiology of Bothrops snakebite, particularly focusing on the interaction between hemostatic and inflammatory mechanisms.

The article incorporates a wide range of sources, including both preclinical and clinical studies, to support its conclusions.

The article proposes potential areas for future research, such as the search for thromboinflammatory markers with predictive or prognostic roles.

The article includes helpful figures and diagrams to aid in understanding the complex interactions between various components of the hemostatic and complement systems.

Weaknesses:

 The article's language is technical and may be difficult for non-experts to understand.

Suggestions for improvement:

 The authors could consider using more accessible language to make the article more approachable for non-experts.

Author Response

# Reviewer 2

Comments and Suggestions for Authors

Strengths:

The article provides a comprehensive review of the current understanding of the pathophysiology of Bothrops snakebite, particularly focusing on the interaction between hemostatic and inflammatory mechanisms.

The article incorporates a wide range of sources, including both preclinical and clinical studies, to support its conclusions.

The article proposes potential areas for future research, such as the search for thromboinflammatory markers with predictive or prognostic roles.

The article includes helpful figures and diagrams to aid in understanding the complex interactions between various components of the hemostatic and complement systems.

Weaknesses:

The article's language is technical and may be difficult for non-experts to understand.

While the article provides a comprehensive review of the literature, it does not include any original research or novel findings.

The article does not discuss the limitations of the studies it cites or address potential confounding factors that may affect the results.

The article does not explore potential differences in the pathophysiology of Bothrops snakebite among different populations or geographic regions.

The article's focus on Bothrops snakebite may limit its applicability to other types of snakebite envenomation.

Questions for the authors:

  1. Have you identified any potential areas of research that you believe could lead to improved treatments for Bothrops snakebite envenomation?

As highlighted between lines 539-543, the greatest need that our review identifies is the lack of biomarkers associated with the clinical complications of Bothrops envenoming, so far, as highlighted is a challenge due to the complex interactions between the blood cascade systems and its cells. Before treatment, there is a need to early diagnose the different clinical complications (kidney damage, stroke, cytokine storm, hemorrhagic manifestations in vital organs, and others). Studies that allow a detailed description of the mechanisms and the definition of the relative contribution of each of these processes to the pathogenesis of different complications, will enable the discovery of specific predictive biomarkers for diagnostic and monitoring purposes. Only then will the development of new complementary drugs to antivenom treatment, and their effectiveness, be evidenced. Currently, our study group is conducting a multicenter clinical study aimed at dissecting the mechanisms associated with severity and clinical complications, as well as identifying biomarkers using different omics tools.

Response:

  1. How does the pathophysiology of Bothrops snakebite compare to that of other types of snakebite envenomation?

Response: Although there is great venomic variability, some clinical manifestations are common among Bothrops envenomations and other species. Such as pain, edema and necrosis, which is also a clinical manifestation in envenoming by Echis snakes, in some cases also by Crotalus snakes. Due to the large number of species, it is impracticable to name them all.

  1. Have you observed any differences in the pathophysiology of Bothrops snakebite among different populations or geographic regions?

Response: As highlighted throughout the article, different toxins from the same family induce unique effects, such as berytractivase and non-hemorrhagic class P-III metalloproteinase from B. erythromelas venom, which acts by activating prothrombin (Line 198). On the other hand, other class P-III metalloproteinases act on platelets (Table 1). In the clinic, hemorrhagic stroke, the ischemic stroke has been described following only B. atrox, B. caribbaeus and B. lanceolatus snakebites (line 231-233), while pulmonary changes - respiratory failure/acute lung edema were reposted only in cases of snakebite by B. atrox (line 173-174). Thus, differences in the mechanism of toxins, and consequently, in the clinic related to the different species are presented in the review. However, due to the absence of clinical trials in different regions of Latin America on Bothrops envenoming, it is difficult to understand the great intra/interspecific diversity of the effects caused by these venoms in humans.

  1. Are there any potential confounding factors that may affect the results of the studies you cite in the article?

Response: The review presents a high number of pre-clinical and clinical articles. Thus, citing their confounding factors would be impracticable. However, an obvious limitation to better understanding the pathophysiology of Bothrops envenoming is undoubtedly the lack of multicentric clinical studies aimed at capturing a wide range of patients with different manifestations and clinical complications. With this, it would be possible to determine the specific mechanisms in an inter/intraspecific way, as well as calculate frequencies, define the peculiarities of Bothrops envenoming cases according to the species, in addition to paving the way for the development of diagnostic, monitoring and therapeutic intervention strategies for according to regional needs. Actually, our research group is currently working on this.

  1. How can your findings be translated into clinical practice to improve outcomes for patients with Bothrops snakebite envenomation?

Suggestions for improvement:

  1. The authors could consider using more accessible language to make the article more approachable for non-experts.

Response: The article has been carefully proofread and corrected.

  1. The authors could include a discussion of potential limitations or confounding factors that may affect the interpretation of the studies they cite.

Response: Vide response of item 3.

  1. The authors could explore potential differences in the pathophysiology of Bothrops snakebite among different populations or geographic regions.

Response: Vide response of item 3.

  1. The authors could consider incorporating original research or novel findings to complement their literature review.

Response: We appreciate the suggestion, however, as it is only a review article, we believe that original data are outside the scope of the proposal. On the other hand, our article opens up the discussion and avenues for the study of thromboinflammation in Bothrops envenoming, which until now has been somewhat under investigated. As presented, the blood systems have been explored in isolation, and here we present an interconnection between them. Thus, it is necessary to develop this area of study as presented in the conclusions of this review.

  1. The authors could broaden their focus beyond Bothrops snakebite to explore the pathophysiology of other types of snakebite envenomation.

Response: Thanks for the suggestion. However, snakes as known, the great variability of clinical effects caused by snakes of different genera and families is wide. Thus, the inclusion of other venoms and snakebite by other species and genus in which the absence of sufficient studies to support the thromboinflammation hypothesis could compromise the quality of the article, leaving only assumptions. In this article, we bring together a collection of experimental and clinical studies that support the establishment of thromboinflammation in Bothrops envenomations, the most relevant genus for envenomations in the Americas.

  1. Overall, I believe that this article is worthy of publication in a peer-reviewed journal, but it could benefit from some revisions and additional analysis.

Response: Thanks for the suggestion. We believe that all questions were clarified, and the modifications made to the manuscripts.

Comments on the Quality of English Language

Weaknesses:

The article's language is technical and may be difficult for non-experts to understand.

Suggestions for improvement:

 The authors could consider using more accessible language to make the article more approachable for non-experts.
